# Effect of Drought and Heavy Metal Contamination on Growth and Photosynthesis of Silver Birch Trees Growing on Post-Industrial Heaps

**DOI:** 10.3390/cells11010053

**Published:** 2021-12-24

**Authors:** Krzysztof Sitko, Magdalena Opała-Owczarek, Gabriela Jemioła, Żaneta Gieroń, Michał Szopiński, Piotr Owczarek, Małgorzata Rudnicka, Eugeniusz Małkowski

**Affiliations:** 1Plant Ecophysiology Team, Institute of Biology, Biotechnology and Environmental Protection, University of Silesia in Katowice, 40-032 Katowice, Poland; gabi.jemiol@gmail.com (G.J.); zgieron@us.edu.pl (Ż.G.); mszopinski@us.edu.pl (M.S.); malgorzata.rudnicka@us.edu.pl (M.R.); eugeniusz.malkowski@us.edu.pl (E.M.); 2Institute of Earth Science, University of Silesia in Katowice, 41-200 Sosnowiec, Poland; magdalena.opala@us.edu.pl; 3Institute of Geography and Regional Development, University of Wrocław, 50-137 Wrocław, Poland; piotr.owczarek@uwr.edu.pl

**Keywords:** *Betula*, birch, photosynthesis, chlorophyll fluorescence, ecophysiology, cadmium, drought

## Abstract

Silver birch trees (*Betula pendula* Roth) are a pioneering species in post-industrial habitats, and have been associated with an expansive breeding strategy and low habitat requirements. We conducted ecophysiological and dendroclimatological studies to check whether there are any features of which the modification enables birch trees to colonise extreme habitats successfully. We characterised the efficiency of the photosynthetic apparatus, the gas exchange, the content of pigments in leaves, and the growth (leaf thickness and tree-ring width) of birch trees on a post-coal mine heap, a post-smelter heap, and a reference site. Birch growth was limited mainly by temperature and water availability during summer, and the leaves of the birch growing on post-industrial heaps were significantly thicker than the reference leaves. Moreover, birch trees growing on heaps were characterised by a significantly higher content of flavonols and anthocyanins in leaves and higher non-photochemical quenching. In addition, birches growing on the post-coal mine heap accumulated a concentration of Mn in their leaves, which is highly toxic for most plant species. Increasing the thickness of leaves, and the content of flavonols and anthocyanins, as well as efficient non-photochemical quenching seem to be important features that improve the colonization of extreme habitats by birches.

## 1. Introduction

Climate change is happening at an accelerating pace—literally before our eyes. Species with high plasticity, such as silver birch (*Betula pendula* Roth), have a great chance of survival, as shown by the species’ pioneers colonising extreme habitats. Plants under stress move from an optimal to a sub-optimal physiological condition and reach a new equilibrium [1]. What changes in growth and physiology make the achievement of this new equilibrium in the post-industrial waste heaps possible?

Silver birch trees are fast-growing and exceptionally tolerant of many environmental events, such as spring frost, low temperatures in general, and nutrient deficiency, although their ecological amplitude is limited by shade intolerance and short life spans [2]. With full leaf development, birch crowns let in up to 37% of total light, allowing many species to grow in the undergrowth while mitigating extreme temperature fluctuations [3]. They are common pioneers in the temperate and boreal forests of Europe but also in fallow lands, post-industrial areas, and highly contaminated sites [2,4,5,6,7]. Silver birch is widely recognised as a species that is resistant to toxic emissions around industrial areas [8,9]. This species occurs in soils contaminated with heavy metals and plays an essential role in the reclamation of heavy metal-rich habitats. Research has shown that birch trees take up and accumulate Cd, Pb and Zn in their organs [8,9]. In addition, they exhibit two types of antioxidant defence systems: those with low molecular weight reagents (phenols) and those with high activity of antioxidant enzymes. For example, the most stressed trees show the highest concentration of polyphenols [6].

With the development of human civilization, anthropogenic pollutants began to play an increasingly important role in the degradation of the natural environment [7]. Hard coal has been mined in Poland since the 17th century [10]. The extraction of 1 ton of coal produces 0.3–0.6 tonnes of coal waste [11]. Coal waste generated during mining activities is usually deposited in nearby heaps, and immediately after depositing, the waste is exposed to weather conditions [12]. Waste mainly consisting of minerals and variable amounts of organic matter begins to weather immediately after settling [12,13]. As a result of a transformation of organic and mineral matter, waste begins to oxidise, which, in some cases, may lead to self-heating and self-ignition of the heap [12,13]. Self-heating leads to the synthesis of new components such as n-alkanes, isoalkanes, alkylcyclohexanes, acyclic isoprenoids, sesquiterpenes, polycyclic aromatic hydrocarbons and phenols [12]. The main mobile and water-soluble compounds formed during self-heating are aniline, quinolone and phenols. They can be washed into groundwater and, as a result, threaten the environment [14]. During self-heating, associated with the transformation of organic and mineral matter, many different gases are released, among which carbon dioxide and carbon monoxide predominate, with additional components such as carbonyl sulphide, carbon disulphide, methane, cyclic alkanes and chloroform, as well as benzene and its alkyl derivatives [11].

Unlike hard coal mining, zinc and lead ores have been mined in Poland since the 12th century. However, the ore processing waste is also stored near the smelters in the form of spoil heaps [15]. The essential components of metal heaps are slag from muffle furnaces, ash, waste from distillation and roasting furnaces, as well as fireclay bricks. Slags from the smelting of no-ferrous metals are toxic, especially under oxidising and well-drained conditions [16]. Weathered material as a whole is chemically more homogeneous than non-weathered slag and has lower concentrations of Si and higher concentrations of Fe, Pb and Cd [17]. The heavy metal content in these heaps is very high and not uniform. Due to the toxicity caused by high concentrations of heavy metals (Zn, Pb and Cd) in the substrate, zinc dumps are the most troublesome [16].

Both presented types of heaps are extreme habitats that are particularly unfavourable for vegetation. In both cases, we are dealing with impeded penetration of the substrate by the roots, lack of access to groundwater, lack of fertility and a very high daily temperature amplitude. On both objects, however, we observe plant succession, and silver birch is always one of the first species. The aim of this work was to conduct a comprehensive study of the ecophysiology of *Betula pendula* trees in extreme habitats such as post-mining and post-smelter heaps compared to the reference area. We established the following research hypotheses:Birches on the control site are characterised by better growth and physiological condition than birches growing on heaps. Many years of growth in extreme habitats should exclude the occurrence of the phenomenon of hormesis [18], while the cumulative toxic effect of environmental factors should translate into a worse physiological condition of trees, including reduced annual growth of wood.There are modifications in the photosynthesis mechanism, enabling the adaptation of birch trees to the extreme conditions of the habitat. Recent scientific work [19,20,21,22] indicates the role of non-photochemical quenching (NPQ) in the adaptation to drought conditions, which is a key stress factor in heaps.It is possible to demonstrate physiological parameters, the modification of which enables birch trees to colonise extreme habitats as one of the first species.

In order to verify the hypotheses, we carried out measurements of selected plant growth parameters, examined their gas exchange, the efficiency of the photosynthetic apparatus, the content of pigments in leaves and the accumulation of selected elements in the tissues, and conducted dendroclimatological analyses. The obtained results made it possible to indicate physiological parameters that may be crucial for the colonization of post-industrial habitats by silver birch trees and their successful life cycle.

## 2. Materials and Methods

### 2.1. Site Description

The study area was located in southern Poland in the Silesian Upland (Figure 1A). This hilly area with maximum elevation c. 400 m a.m.s.l. is built of sedimentary Paleozoic and Mesozoic rocks containing valuable coal, iron, zinc and lead deposits (the Upper Silesia Coal Basin). These mineral resources influenced the development of Poland’s most important industrial region, and the landscape of Upper Silesia is highly urbanised. The Upper Silesian Industrial Region covers 3200 km² and is inhabited by c. 3 million people (Figure 1B). The densely populated area centred around Katowice with mining, iron and steel (iron and nonferrous metals) industrial activity is also one of the most polluted areas not only in Poland, but also in Europe [23,24]. Coal mining in the Upper Silesia commenced as early as the end of the 18th century, but even earlier, in the late Middle Ages, the mining of lead and silver began. Intensive human activity led not only to soil and air pollution, but also to the relief and surface and ground water network transformation [25,26]. One of the most characteristic elements of relief documenting industrial activity and its influence on the natural environment are numerous dumps and heaps, among which two research areas were selected for sample collection. (Figure 1B). The lead–zinc ore spoil heap is located in Piekary Śląskie (‘P’) in the northern part of the Upper Silesian Industrial Region. This oval shape heap with an area of 4.1ha and a height of 20–25 m is located at 273 m a.m.s.l. (Figure 1C). The second research area includes the post-coal mine spoil heap located in Katowice-Murcki (‘K’) at 304 m a.m.s.l. (Figure 1D). It is a vast, elongated heap with an area of 14,1 ha and a height of c. 30 m. Additionally, a reference area (‘T’) was designated in the southern part of the Upper Silesian Industrial Region in the vicinity of Tychy. The physicochemical characteristics of the soil and ecological data for the studied sites were additionally presented in Appendix A.

### 2.2. Plant Material and Sampling

Ten individual birch trees from each population (‘T’, ‘K’, and ‘P’) were selected for measurements. Single-trunk, free-standing trees were selected, with no obvious tumour lesions on the trunk, scars and fungi. The birch populations in the heaps were age homogeneous and did not show signs of vegetative reproduction. Trees that were as free-standing as possible, without root suckers, and significantly distant from other individuals were selected for the research. Soil samples were taken directly from under each tree for further analysis. All in situ measurements of physiological parameters were performed between June 25 and 30, in 2020. The selected trees were biological replicates, and the following number of in situ measurements was performed on the selected leaves for each tree: fifteen measurements of content of leaves’ pigments using a pigment content meter (Dualex Scientific+, Force-A, Orsay, France); four measurements of non-photochemical quenching using a PAM fluorimeter (MultispeQ, PhotosynQ, East Lansing, MI, USA); five measurements of chlorophyll *a* fluorescence using a fluorimeter (Pocket PEA, Hansatech Ltd., Pentney, UK); ten measurements of gas exchange parameters using an infra-red gas analyser (Targas 1.0, PP-Systems, Amesbury, MA, USA). The measurement methodology was in line with published articles [27,28]. The measurements of pigment content and chlorophyll fluorescence were performed on the same leaves. A pigment content meter (Dualex Scientific+, Force-A, Orsay, France) was used to measure the chlorophyll, flavonol and anthocyanin content in leaves. Measurements were taken after the device was automatically calibrated by placing a leaf blade between the measuring heads. Care was taken to measure the leaf surface without major veins. Chlorophyll *a* fluorescence measurements were conducted for the same plants using the Plant Efficiency Analyser (PocketPEA fluorimeter; Hansatech Ltd., Pentney, UK). Before being measured, each selected leaf was adapted in the dark for 30 min using leaf clips. After adaptation, a saturating light pulse of 3500 mmol quanta m^−2^ s^−1^ was applied for 1 s, which closed all of the reaction centres, and the fluorescence parameters were measured. Pulse-amplitude-modulated fluorescence was measured using a PAM fluorimeter (MultispeQ, PhotosynQ, East Lansing, MI, USA) and the Photosynthesis RIDES protocol. Both chlorophyll content and fluorescence measurements were performed in situ without destroying the plant material (ten plants per site were examined; fifteen measurements of pigment content per plant, four measurements of PAM fluorescence per plant and five measurements of chlorophyll fluorescence for JIP-test per plant were taken). Gas exchange measurements were performed on the same leaves as measurements of chlorophyll fluorescence using an infra-red gas analyser (Targas 1.0, PP-Systems, Amesbury, MA, USA). The examined leaves were placed in the measuring chamber of the device, and two measurements were made on each of the five leaves for each tree after waiting for the photosynthesis and transpiration rate readings to stabilise (about 2 min). The investigated leaves were collected, transported to the laboratory, prewashed thoroughly in tap water and then washed using an ultrasound washer filled with deionised water. Subsequently, the plant samples were dried at 80 °C for 72 h. Then, the dry plant material was ground in a mortar, mixed and acid digested.

For each birch tree, three samples of the soil from its immediate surroundings were collected and mixed together in order to obtain one composite sample per plant. In the laboratory, the soil samples were air-dried and sieved through a 2 mm screen before measuring their pH and EC and analysing their mineral compositions. Soil pH was measured in deionised water (1:2.5, m/vol) and 1 M KCl (1:2.5, m/vol) using a combination glass/calomel electrode (OSH 10–10; Elmetron, Zabrze, Poland) and a pH/conductivity metre (CPC-505; Elmetron, Zabrze, Poland) at room temperature after 24 h of equilibration. The electrical conductivity was determined in a deionised water suspension (soil:solution ratio of 1:2.5, m/vol) at room temperature after 24 h of equilibration using a glass conductivity cell (EC-60; Elmetron, Zabrze, Poland) and a pH/conductivity metre (CPC-505; Elmetron, Zabrze, Poland).

The plant and soil material was acid-digested in a microwave-assisted wet digestion system (ETHOS 1, Milestone, Sorisole, Italy) according to the procedure provided by the manufacturer (concentrated HNO_3_ and H_2_O_2_, 4:1 vol/vol). The concentration of metals was analysed in the extracts (soil, CaCl2) and digests (plant, soil) using flame atomic absorption spectrophotometry (iCE 3500 FAAS, Thermo Fisher Scientific, Waltham, MA, USA). A reference plant (Oriental Basma Tobacco Leaves (INCT-OBTL-5), Institute of Nuclear Chemistry and Technology, Warszawa, Poland) and soil material (NCS DC 77302, China 502 National Analysis Center for Iron and Steel, Beijing, China) were used to determine the quality assurance of the analytical data.

Core samples were collected for each tree using Pressler’s borer from two spoil heaps and the reference site in 2019 and 2020. Samples were processed using standard dendrochronological procedures [29]. Tree-ring widths (TRW) were measured with an accuracy of 1/100 mm using the WinDENDRO measuring system; then, sequences were dated and crosschecked with the Cofecha program [30]. For all sites, the Expressed Population Statistics (EPS), an indicator of chronology reliability, remained above 0.85 (a generally accepted cut-off). Site chronologies were constructed from well-correlated samples (r > 0.4). The highest correlation between sequences was calculated for the Piekary Śląskie site, but only 7 out of 10 samples were correctly dated at this site. All 10 samples were successfully cross-dated at the Katowice Murcki site; the average series intercorrelation among samples at this site was 0.492 and the mean sensitivity was 0.622. The ring-width chronologies of silver birch from the Upper Silesian Industrial Region spanned the common period of 2001–2019, i.e., 18 years, with an average radial growth rate from 2.640 to 4.250 mm/year (Table 1).

### 2.3. Statistical Analysis

The results are shown as the means ± SE. Statistically significant differences among the mean values were determined using a one-way ANOVA and a post hoc Tukey HSD test (*p* < 0.05). The statistical analysis was performed using Statistica v.13.1 software (Dell Inc., Austin, TX, USA). The pipeline models of energy fluxes through leaf ’s cross section were done using CorelDRAWX6 (Corel Corp., Ottawa, ON, Canada).

## 3. Results

### 3.1. Growth Parameters

The growth of plants was characterised by measurements of mean tree-ring width from the last five years (WTR_5_) and throughout the entire life of the tree (WTR_T_) (Table 1). The widest tree-rings were measured in birches from the post-coal mine spoil heap (K). The narrowest tree rings were observed in trees growing on the lead–zinc ore spoil heap (P). Moreover, significant differences were measured in the thickness of tree leaves in the studied sites (Table 1). The thinnest leaves were observed at the reference site (T). The birches in Piekary (P) had leaves that were twice as thick as those at the reference site, while leaves that were three times thicker than those in Tychy were observed in Katowice (K). Additionally, pictures of typical leaves for all investigated populations were taken (Appendix A). There were traces of phytophagous insects feeding on leaves from the K population and clear chlorosis in leaves from the P population. It is worth noting that the best growth was recorded for birches growing on the post-coal mine spoil heap (K), which is an extremely poor habitat, without access to groundwater and exposed to many stress factors.

### 3.2. Pigment Content and Gas Exchange Parameters

Reference plants were characterised by the highest chlorophyll content in their leaves (Table 1). A significantly lower value was observed in leaves of ‘K’ trees, whereas ‘P’ plants had the lowest chlorophyll content in their leaves. On the other hand, trees from both post-industrial heaps showed significantly higher anthocyanin content in the leaf epidermis than reference plants (Table 1). The highest content of flavonols was measured in ‘K’ tree leaves. Additionally, in ‘P’ leaves, the flavonols content was 20% higher than in the reference population (Table 1). It seems that flavonols, known for their antioxidant properties, may play a crucial role in adaptation to extreme habitat conditions.

The rate of CO_2_ assimilation was the highest in the reference plant leaves, with the lowest measured intracellular CO_2_ concentration at the same time (Table 1). The opposite relation was observed in ‘P’ plants. Trees from Katowice were characterised by middle values, significantly different from both ‘T’ and ‘P’ plants (Table 1). Changes in transpiration rate showed the same relation as the chlorophyll content and photosynthetic rate between the investigated populations. At the same time, no differences were found in stomatal conductance among plants from the tested sites (Table 1).

### 3.3. Element Accumulation in Plant Tissues

In the whole ionomics study, the most important finding seemed to be the manganese content in the leaves of birch trees in population ‘K’, which was over ten times higher than that of the control and the population of P (Table 1). Significantly higher Mn contents were also observed in the wood and bark of the ‘K’ population, which excluded surface contamination of the leaves (Appendix A). In addition, manganese accumulation increased in older wood increments, which proves the multi-year scale of the process of Mn accumulation in the tissues of the population of ‘K’ (Appendix A). A similar increase in accumulation in wood with age was observed for all populations in the case of zinc (Appendix A). It is worth noting that the ‘P’ population was the only one characterised by the presence of Cd in leaves, and the highest levels of Ca, Mg and Zn, both in leaves (Table 1), bark and wood (Appendix A).

### 3.4. Chlorophyll Fluorescence Characteristics

The reference population ‘T’ was characterised by the lowest minimum fluorescence and, at the same time, the highest maximum fluorescence among the studied populations, which proves that this population was correctly selected as a control (Figure 2A). At the same time, the shape of the curves indicates that the efficiency of the photosynthetic apparatus in both populations of birch trees inhabiting the heaps was lower than that of the reference trees (Figure 2A). Additionally, the potential damage was more related to PSII in ‘K’ birches, and PSI could be more damaged in ‘P’ trees. Birches from both types of heaps were characterised by the presence of peaks ΔI, ΔJ and ΔG, which could indicate a decrease in—or damage to—the ubiquinone pool, an accumulation of Q_A_^−^ (reduced primary quinone) and decrease in activity of the final electron acceptors in PSI, respectively (Figure 2B). Moreover, the ΔI and ΔJ peaks were significantly higher for ‘K’ trees when compared to ‘P’ trees. The inverse relationship was observed for the peak ΔG. The presence of all the peaks in ‘P’ birches may be related to the accumulation of the toxic effect of drought and heavy metals, which have a destructive effect on the photosynthetic apparatus. These dependencies had their consequences in the form of changes in the detailed parameters characterising the activity of photosystems, which will be discussed below.

Both populations of birches from heaps were characterised by significantly lower values of the maximum quantum yield of the primary PSII photochemistry (φP_0_), and it was probable that a trapped exciton moved an electron into the electron transport chain beyond Q_A_^–^ (ψE_0_) and the quantum yield for electron transport from Q_A_^−^ to plastoquinone (φE_0_) as compared to the reference trees (Figure 3). At the same time, population ‘K’ had significantly lower values of these parameters than population ‘P’. Moreover, both heaps’ populations had higher quantum yields of energy dissipation (φD_0_) compared to the reference trees. Only birches from the coal heap had significantly higher values of the probability with which an electron from the intersystem electron carriers would move to reduce the end acceptors at the PSI acceptor side (δR_0_). Importantly, the ‘K’ population had significantly higher values of non-photochemical quenching (NPQ_T_ and φNPQ) compared to the other studied populations.

Birch trees from both the ‘K’ and ‘P’ heaps were characterised by lower values of energy absorbed and trapped by photosystem II as compared to reference site ‘T’ (Figure 4). Moreover, their electron transport flux and percentage of active reaction centres were comparable, and they were both significantly lower compared to those of the ‘T’ trees. The highest dissipated energy flux (DI/CS) was measured for the ‘K’ plants (Figure 4), which somehow confirms the high ΔI and ΔJ peaks (Figure 1B).

### 3.5. Dendroclimatological Analysis

Tree-ring width (TRW) data were calibrated against meteorological data from Katowice meteorological station (1951–2020) to assess the climate–growth relationships at sites with varying degrees of contamination (Figure 5). Due to the short time series, the discussed results are statistically significant at *p* < 0.01. Correlation analysis indicated that *B. pendula* growth is limited mainly by temperature and water availability during summer (‘T’ site, Figure 5). Dendroclimatological analysis for August, showing negative correlation with temperature and positive correlation with precipitation, indicated possible drought stress during late summer. The highest correlation was obtained with the May temperatures for all investigated sites. Dendroclimatic analysis revealed the strongest May temperature signal in the ‘T’ site (r = −0.50), and a weaker but still significant signal at the ‘K’ (r = −0.42) and ‘P’ (r = −0.42) sites. Dendroclimatological analysis clearly showed that the growth rings of birch from contaminated sites are less sensitive to climate variability. This was found to be especially evident for the most polluted site ‘P’, as the radial growth of birches from the ‘K’ site showed positive relationships with the June–July mean temperatures (r = 0.42). Furthermore, the results obtained for precipitation conditions indicate some importance of precipitation in December (‘T’ trees) and April (‘P’ trees). However, relationships with regional snow cover extent require further research. In particular, unfavourable periods for growth, so-called negative indicator years, were distinguished in all investigated site chronologies in 2007, 2013 and 2018. The above years were characterised by a very high number of hot days (>58 days), especially in 2018 (80 days), which was an exceptional year in the whole temperature series for Katowice in this respect.

### 3.6. Determining the Correctness of Selection and Correlation between the Tested Parameters

The presented analysis of the main components made it possible to characterise almost 65% of the variability between the studied populations (Figure 6). All the studied populations formed separate groups, so we can conclude that the selected research methods allowed the capturing of the variability of the studied populations. At the same time, on the basis of the correlation matrix, we can conclude that the largest increases in the trunk thickness of Katowice birch trees were positively correlated with the increase in leaf thickness, NPQ and manganese accumulation in leaves (Appendix A).

## 4. Discussion

In the present research, we put forward a hypothesis that birch trees in the control habitat would be characterised by a better growth and physiology compared to trees inhabiting the post-industrial heaps. Surprisingly, it turned out that birches from the extreme habitat of a mine waste heap were characterised by the widest tree-rings and the thickest leaves (three times thicker than the reference leaves) amongst tested populations (Table 1). These surprising results can be related to the drought and salinity occurring in the mine rock heap. Kitao et al. [31] showed that birch trees subjected to drought stress are characterised by significantly thicker leaves than those watered daily, which is one of the features of long-term drought-acclimation. It is worth noting that the research was carried out on two-year-old seedlings growing in pots [31], and our results were obtained for at least fifteen-year-old trees in situ. Moreover, the results of dendroclimatological research obtained for birch from the area of Upper Silesia, showing mainly the influence of summer temperatures and drought stress, are slightly different than for pine and oaks, for which the highest dependence on winter temperature was obtained, as well as the second-highest dependence on rainfall conditions in summer [32,33]. On the other hand, it was found that the birches on the heaps are characterised by lower biometric parameters and reproductive abilities than the reference trees [34]. Additionally, birch trees exposed to heavy metals had leaves that were twice as thick as those of the reference trees (Table 1). So far, it has been shown that in metalliferous heaps, plants have thinner leaves than in control habitats, e.g., *Biscutella laevigata* [35]. We can speculate that both thicker leaves and the trunk may be part of a strategy that allows birch trees to survive in an extremely unfavourable habitat where every volume of water is essential, available only for a short time immediately after rainfall.

The results described in this study showed a strong positive correlation between the accumulation of manganese in birch leaves and their growth (Figure 6; Appendix A). On the one hand, Mn is a micronutrient necessary for plants’ proper growth and development [36,37]. On the other hand, its accumulation exceeding 150 mg g^−1^ DW causes a toxic effect and disrupts many physiological processes [37]. Reimann et al. [38] showed that Mn is easily available for birches in soils with low pH, and its accumulation in leaves under such conditions, even at values up to 4900 mg g^−1^ DW, is possible. Moreover, Mn is either immobilised by the high pH or its uptake is blocked by the high Ca concentration [38], which may explain the very low content of Mn in the ‘P’ birch tissues, despite the high content of Mn in the heap substrate. Kitao et al. [39], conducting hydroponic experiments on two-year-old seedlings, proved that the accumulation of Mn at a concentration of 20 ppm in white birch leaves induced chlorosis and brown speckles. The birch leaves from the ‘K’ heap, apart from the traces of insects’ feeding, were not characterised by any visible signs of toxicity, which demonstrates an efficient mechanism of detoxification of high concentrations of Mn (>3000 μg g^−1^ DW; Table 1) and of adaptation of birch trees to the harsh habitat. We can only speculate as to whether the mechanism of this adaptation is directly related to better growth. The topic definitely requires further research.

It was claimed in the literature that the heavy metal content in birch leaves is similar to that obtained from other vegetation species from Pb/Zn mining areas [5]. We showed that birch trees on heap ‘P’, heavily contaminated with HMs, accumulated a very low concentration of Cd in the leaves and a concentration of Zn that did not exceed 600 mg g^−1^ DW (Table 1). There are articles that provide data on birch trees accumulating significantly higher levels of heavy metals in their leaves, with Zn accumulation exceeding 3000 ppm [5,6]. The previous research on the influence of air pollution on the dendroclimatic signal in pines from Upper Silesia confirmed the general tendency for the climatic signal embedded in tree rings from the polluted sites to weaken [33].

Most of the studies on spoil heaps showed that the plants growing on them have a lower chlorophyll content than plants from reference habitats [40,41,42]. Birch trees have been no exception so far [6], and our research also confirms this fact (Table 1). Chlorophyll fluorescence parameters are very often used in monitoring plant responses to drought [31,43,44,45], salinity [45] and heavy metals [40,41,46,47,48], i.e., the main abiotic stress factors present in the heaps of our choice. A characteristic plant reaction to the above stresses is an increase in the value of dissipation energy and a decrease in the efficiency of energy fluxes, as well as in its quantum efficiency, through photosystems [28,40,41,45,46,47]. Our research confirmed the above relationships, although it also showed another interesting trait of the ‘K’ population related to the fact or their almost-twice-as-high non-photochemical quenching compared with the reference plants (Figure 3). So far, it has been shown that drought tends to cause a decrease in NPQ values in birch [31,43]. The studies on beech showed, inter alia, a slight increase in NPQ during drought, although this was rather insignificant [48]. Pšidová et al. [44] showed an increase in NPQ during drought in one of the tested beech populations; unfortunately, they did not discuss this observation. The habitat of the mentioned population was the driest and the warmest among those tested by the authors [44]. The results we obtained allow the efficient NPQ protection mechanism to be included in the above-discussed mechanisms, thereby enabling the birches from the ‘K’ heap to survive in extremely unfavourable conditions. The latest research on arable crops also indicates NPQ as a key parameter in the plant response to drought stress [22,49,50]. In addition, Bussotti et al. [51] showed that, in ecological studies, the most important parameters of chlorophyll fluorescence are φP_0_ and ΔV_I–P_. The presented results confirmed that the trees growing on both types of heaps differed significantly between each other and compared to the control, both in the values of the φP_0_ (Figure 3) parameter and the fluorescence curves on the I–P section (Figure 2B).

Quite a few studies have shown the negative effect of drought [2,44] and heavy metals [49,50] on gas exchange in trees, although they were run under controlled conditions and pot experiments. Our results confirmed that drought and high temperatures, which were present in both types of the studied heaps, significantly reduced photosynthesis and transpiration. Additionally, heavy metals that were present in the ‘P’ substrate and accumulated in leaves further reduced the gas exchange parameters in birches (Table 1). The presented results, in the context of both the selected research sites and the comparison of ecophysiological and dendroclimatological analyses, are new for science.

## 5. Conclusions

The presented ecophysiological studies aimed to characterise the birch features that enable the colonization of extreme habitats. Post-industrial waste heaps are extremely difficult to live in for plants due to a lack of access to groundwater, high daily and annual temperature amplitudes, deficiencies in macro- and microelements, poor soil structure, and the presence of toxic chemicals including heavy metals. So far, scientific data have indicated that an expansive breeding strategy and low habitat requirements are behind the birch’s success. Based on our research, we can speculate that one of the mechanisms enabling the effective colonization of extreme habitats by birch trees is the increasing of the thickness of the leaves and the annual growth rate of the wood in order to store water. In addition, an important role seems to be played by the efficient protective mechanisms of the photosynthetic apparatus, such as non-photochemical quenching, and increased production of anthocyanins and flavonols in leaves. Such improvements in photosynthesis could be crucial in the context of climate change, as drought and high temperature amplitudes become commonplace, just as they are currently the norm on post-industrial heaps.

The issue of high Mn accumulation in plant tissues of birches from post-coal mine heaps may result from the specificity of the habitat (low pH of substrate). However, we recommend conducting more extensive research on this issue because tolerance to such high concentrations of this microelement and its mechanism are worth exploring. We also recommend conducting research on tree adaptation strategies, especially in the era of rapid climate change. In situ monitoring of physiological parameters seems advisable in this context, although modelling in laboratory conditions may also be effective.

## Figures and Tables

**Figure 1 cells-11-00053-f001:**
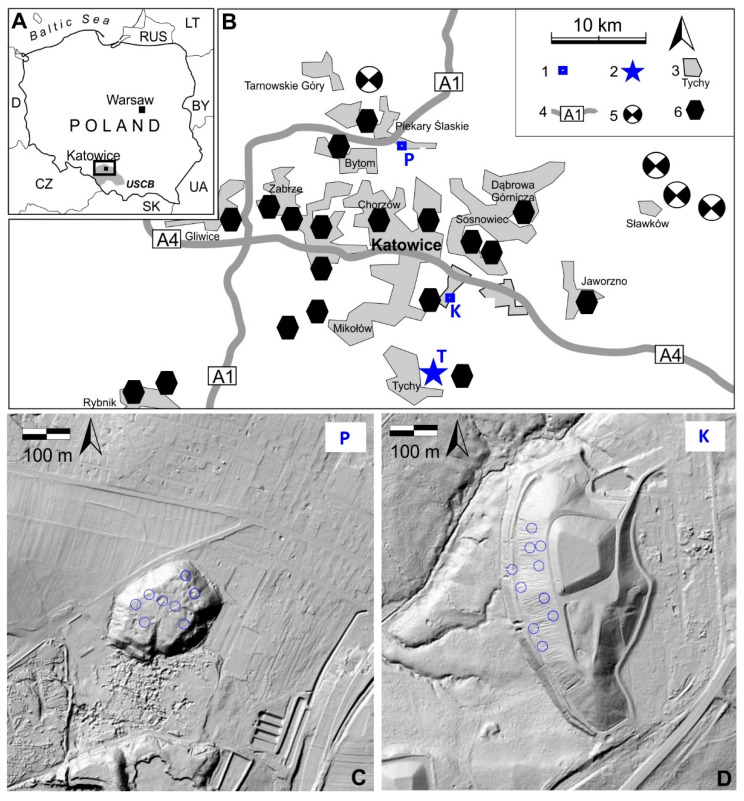
Location of the study area in the southern part of Poland (**A**), a schematic map of the Upper Silesia Industrial Region (**B**): 1—location of research sites (K—Katowice Murcki; P—Piekary Śląskie); 2—location of reference site; 3—urbanised area; 4—main highways; 5—location of old zinc and lead mines; 6—location of main coal mines, both old and in operation. Morphology of lead–zinc ore spoil heap in Piekary Śląskie (**C**), and post-coal mine spoil heap in Katowice (**D**) with location of sampling sites (blue circles). Abbreviations: T—reference population from Tychy; K—birch population from post-coal mine spoil heap; P—birch population from post-smelter spoil heap.

**Figure 2 cells-11-00053-f002:**
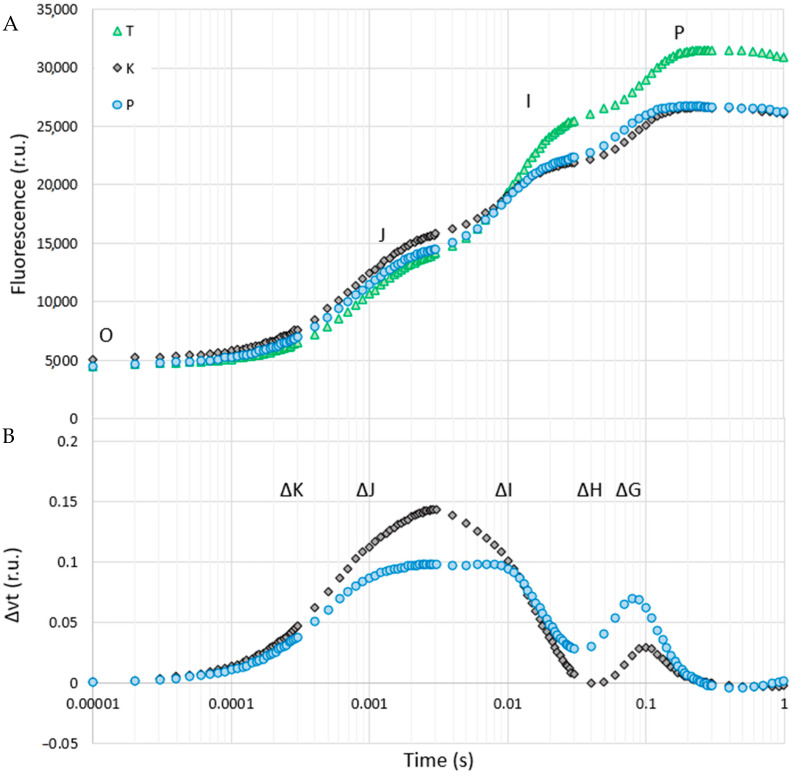
Chlorophyll a fluorescence induction curves of the silver birch (*Betula pendula*) from various sites (**A**) and differences between the populations studied on the relative variable fluorescence (ΔVt = ((Ft − F_0_)/Fv) − Vt_control) (**B**). For ΔVt analysis, the fluorescence of trees from Tychy was the reference and was equal to 0. Values are means (n = 10). Abbreviations: T—reference population from Tychy; K—birch population from post-coal mine spoil heap; P—birch population from post-smelter spoil heap; O, J, I, P—points characterising the course of the fluorescence curve, including the minimum and maximum chlorophyll fluorescence for a given population. ΔK—peak informing about the potential damage of the Oxygen-Evolving Complex; ΔJ—peak describing the level of potential damage of the ubiquinone pool; ΔI—peak corresponding to the accumulation of reduced primary quinone; ΔH and ΔG—peaks describing a decrease in activity of the final electron acceptors in PSI, r.u.—relative units.

**Figure 3 cells-11-00053-f003:**
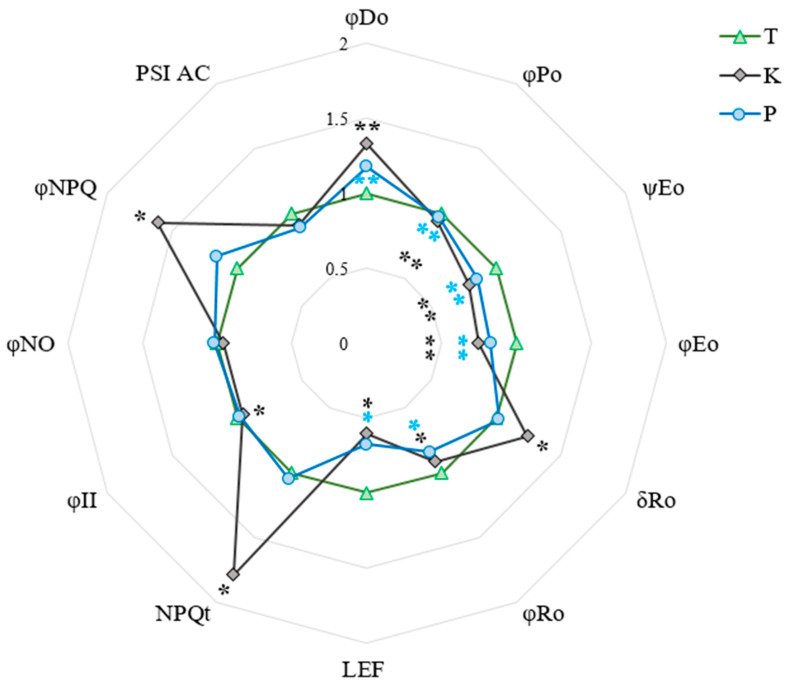
Changes in parameters characterising the state and efficiency of the photosynthetic apparatus of birch trees under the influence of environmental factors at investigated sites. The data presented are means (n = 10). Means followed by single asterisks are significantly different from the reference, whereas those with double asterisks differ from both the reference and the other population, based on the Tukey HSD test (*p* ≤ 0.05). Abbreviations: T—reference population from Tychy; K—birch population from post-coal mine spoil heap; P—birch population from post-smelter spoil heap; φP_0_—maximum quantum yield of the primary PSII photochemistry; ψE_0_—probability (at time 0) that a trapped exciton moves an electron into the electron transport chain beyond Q_A_^–^; φE_0_—quantum yield for electron transport from Q_A_^−^ to plastoquinone; δR_0_—probability with which an electron from the intersystem electron carriers will move to reduce the end acceptors at the PSI acceptor side; φR_0_—quantum yield for the reduction in the end electron acceptors at the PSI acceptor side; φD_0_—quantum yield (at t = 0) of energy dissipation; NPQ_T_ and φNPQ—non-photochemical quenching; LEF—linear electron flow; φNO—non-regulatory energy dissipation; φ_II_—quantum yield of photosystem II; PSI AC—active centres of photosystem I.

**Figure 4 cells-11-00053-f004:**
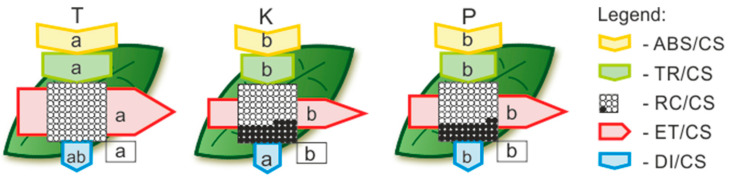
Leaf model showing the phenomenological energy fluxes per the excited cross sections (CS) of the leaves of the B. pendula growing on the reference site (T), post-coal mine heap (K), and post-smelter heap (P). Each relative value of the measured parameters is the mean (n = 10) and the width of each arrow corresponds to the intensity of the flux. Yellow arrow—ABS/CS, absorption flux per CS (approximated); green arrow—TR/CS, trapped energy flux per CS; red arrow—ET/CS, electron transport flux per CS; blue arrow—DI/CS, dissipated energy flux per CS; circles inscribed in squares—RC/CS, % of active/inactive reaction centres. White circles inscribed in squares represent reduced Q_A_ reaction centres (active), black circles represent non-reduced Q_A_ reaction centres (inactive); 100% of the active reaction centres responded with the highest mean value observed in the control conditions. Means followed by the same letter for each parameter in a row are not significantly different from each other according to the Tukey HSD test (*p* < 0.05). Letters are inscribed into arrows, except for RC/CS, where they are placed in a box in the bottom right corner of the square with circles. Abbreviations: T—reference population from Tychy; K—birch population from post-coal mine spoil heap; P—birch population from post-smelter spoil heap.

**Figure 5 cells-11-00053-f005:**
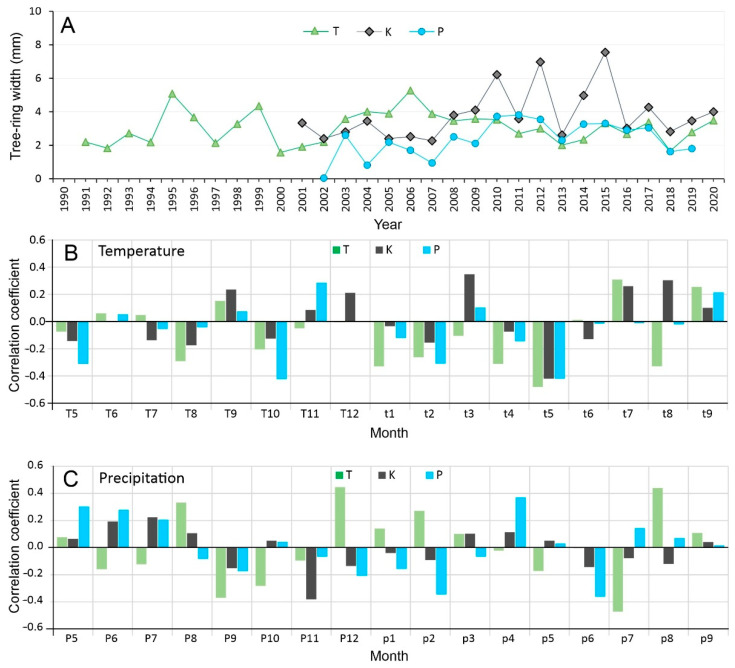
Tree-ring chronologies of *Betula pendula* from three investigated sites in southern Poland (**A**), with their dendroclimatic responses in terms of mean monthly temperatures (**B**) and monthly precipitation totals (**C**) recorded at Katowice meteorological station. The letters on the x-axis represent months from May of the previous year to September of the current year. Abbreviations: T—reference population from Tychy; K—birch population from post-coal mine spoil heap; P—birch population from post-smelter spoil heap.

**Figure 6 cells-11-00053-f006:**
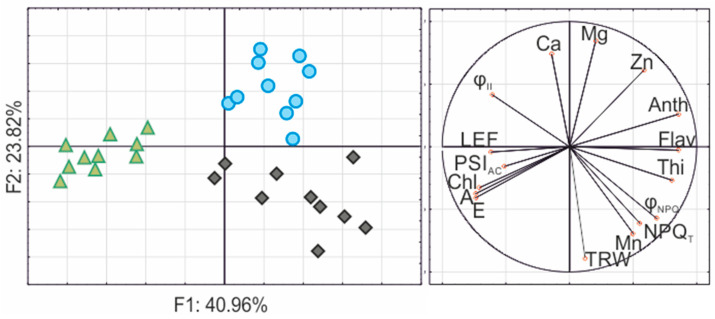
The Principal Component Analysis (PCA) presents the relationships between selected photosynthetic parameters, pigment content and element content of the silver birch from various sites. Legend: green triangle—reference population from Tychy; black rhombus—birch population from post-coal mine spoil heap; blue circle—birch population from post-smelter spoil heap. Abbreviations: Ca—calcium concentration in leaves; Mg—magnesium concentration in leaves; Mn—manganese concentration in leaves; Zn—zinc concentration in leaves; TRW—mean tree-ring width throughout the life of the tree; A—photosynthetic rate; E—transpiration rate; Thi—leaf thickness; Anth—anthocyanin content; Chl—chlorophyll content; Flav—flavonol content; NPQ_T_ and φ_NPQ_—non-photochemical quenching; LEF—linear electron flow; φ_II_—quantum yield of photosystem II; PSI _AC_—active centres of photosystem I.

**Table 1 cells-11-00053-t001:** Plant growth, gas exchange, pigment content and element accumulation in leaves.

	T	K	P
Growth and Physiological Characteristics			
Leaf thickness (mm)	0.08 ± 0.01 c	0.31 ± 0.01 a	0.20 ± 0.03 b
TRW_5_ (mm)	3.10 ± 0.b	3.77 ± 0.a	2.21 ± 0.c
TRW_T_ (mm)	3.06 ± 0.b	3.83 ± 0.a	2.35 ± 0.c
Chlorophyll content (r.u.)	35.6 ± 0.9 a	31.04 ± 1.25 b	28.22 ± 0.71 c
Flavonol content (r.u.)	1.38 ± 0.04 c	1.76 ± 0.02 a	1.67 ± 0.03 b
Anthocyanin content (r.u.)	0.110 ± 0.002 b	0.137 ± 0.005 a	0.141 ± 0.003 a
A (μmol CO_2_ m^−2^ s^−1^)	13.8 ± 0.4 a	11.0 ± 0.5 b	9.5 ± 0.3 c
C_i_ (μmol CO_2_ mol^−1^)	244 ± 5 b	257 ± 3 b	280 ± 5 a
E (mmol H_2_O m^−2^ s^−1^)	2.98 ± 0.15 a	2.20 ± 0.07 b	1.81 ± 0.05 c
g_s_ (μmol H_2_O m^−2^ s^−1^)	205 ± 13 a	208 ± 13 a	202 ± 7 a
Element Accumulation in Leaves (μg g^−1^ DW)			
Ca	10,200 ± 560 a	7520 ± 610 b	11,580 ± 1050 a
Cd	BDL	BDL	1.43 ± 0.19 a
Cu	5.4 ± 0.4 a	3.0 ± 0.4 b	2.0 ± 0.3 b
Fe	86 ± 5 a	82 ± 6 a	89 ± 6 a
K	5960 ± 460 a	4100 ± 420 b	3720 ± 270 b
Mg	2200 ± 110 c	2910 ± 180 b	3980 ± 180 a
Mn	290 ± 90 b	3180 ± 130 a	110 ± 10 b
Zn	195 ± 27 b	148 ± 8 b	550 ± 61 a

The data presented are means ± SE. Means followed by the same letter in a row are not significantly different from each other according to the Tukey HSD test (*p* ≤ 0.05). Abbreviations: T—reference population from Tychy; K—birch population from post-coal mine spoil heap; P—birch population from post-smelter spoil heap; BDL—below detection limit; TRW_5_—mean tree-ring from last five years; TRW_T_—mean tree-ring width throughout the life of the tree; A—photosynthetic rate; c_i_—intracellular CO_2_ concentration; E—transpiration rate; g_s_—stomatal conductance, r.u.—relative units.

## Data Availability

The data presented in this study are available on request from the corresponding author.

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
