# Peer review of "Effect of Drought and Heavy Metal Contamination on Growth and Photosynthesis of Silver Birch Trees Growing on Post-Industrial Heaps"

_cells, 2021, doi:10.3390/cells11010053_

Round 1

Reviewer 1 Report

Considering that birch reproduces vegetatively (especially in unfavorable conditions) how did the authors ensure that selected species are not clones. This should be added to the material and methods section.

Some minor edits are necessary in the text of the English language. there should be more discussion about the scientific contribution of the paper. 

lp

Line 123: to measurements to be replaced with: for measurements

Author Response

Comment 1: Considering that birch reproduces vegetatively (especially in unfavorable conditions) how did the authors ensure that selected species are not clones. This should be added to the material and methods section.

Response: We would like to thank the Reviewer for paying attention to an important detail. We chose relatively young heaps for the experiment to investigate the early stages of succession. Our dendrochronological research shows that the examined birches were about 20 years old. Being in the field, we tried to select young trees of similar shape and size, at the same time significantly distant from each other. We have not found the presence of older trunks, bio-groups, or groups of trees with conical growth characteristics so characteristic of vegetative reproduction. On the other hand, the substrate on heaps, unlike, for example, rock rubble in the mountains, is extremely unfavourable for root penetration. In conclusion, we did not notice that the studied heaps had older trees, the clones of which could have been trees selected for the study. We also did not notice any individual tree possessing the characteristics of vegetative reproduction, perhaps due to the structure of the substrate. We have added relevant information to the M&M section as suggested by the Reviewer (See lines 139-142 in improved MS).

Comment 2: Some minor edits are necessary in the text of the English language. there should be more discussion about the scientific contribution of the paper. 

Response: We would like to thank the reviewer for his comment. The discussion was expanded. The manuscript has been revised by a philologist.

Comment 3: Line 123: to measurements to be replaced with: for measurements

Response: The correction was introduced as suggested by the Reviewer (See line 137).

Reviewer 2 Report

Nowadays, this topic is a key issue in modern society and is receiving great attention around the world. The article entitled "Effect of drought and heavy metal contamination on growth and photosynthesis of silver birch trees growing on post-industrial heaps" is an interesting study that seeks to serve as a basis for future work.

The document is generally clear and well written. The manuscript is prepared in good order with detailed data. On the other hand, the relevance of the review is well reflected. English does not present major problems, it is well written and understandable. In addition, the results are appropriately interpreted and significant. As for the technical design, it is well designed and the data are sufficiently robust to draw its conclusions in such a way that I RECOMMEND THEIR PUBLICATION in Cells Journal.

Author Response

Thank you for your favourable opinion on our manuscript.

Reviewer 3 Report

Review Comments

In this manuscript, the authors conducted ecophysiological and dendroclimatological studies to check whether there are any features, and modification of which enables birch trees to colonize extreme habitats successfully. Birch growth was controlled mainly by temperature and water availability during summer, and the leaves of the birch inhabited post-industrial heaps were significantly thicker than the reference leaves. Moreover, birch trees growing on heaps were characterized by a significantly higher content of flavonols and anthocyanins in leaves and higher non-photochemical quenching. In addition, birches growing on the post-coal mine heap accumulated the concentration of Mn in their leaves, which is highly toxic for most plant species.

Increasing the thickness of the leaves, the content of flavonols and anthocyanins, and efficient non-photochemical quenching seem to be important features that improve the colonization of extreme habitats.

- The study has been well conducted. However, some revisions are still required as shown below;

- The introduction is somewhat well written, but the authors should add a separate section in the Introduction section discussing the research hypothesis in detail and you may also include some more recent literature highlighting that.

- Plant sampling is well-explained in the Methods section, but the authors should also write the methods used in more detail to be reproducible. How many replicates the authors used?

- In Results section, please explain Figure 2 “Chlorophyll a fluorescence induction curves” further and highlight its significance “as the current explanation is too short!”

- Discussion should be interpreted with the results of this work and previous work “comparison” agreement or disagreement” in more details too.

- Conclusion section should also include the concluded recommendation for future work. Adjust that.

Author Response

In this manuscript, the authors conducted ecophysiological and dendroclimatological studies to check whether there are any features, and modification of which enables birch trees to colonize extreme habitats successfully. Birch growth was controlled mainly by temperature and water availability during summer, and the leaves of the birch inhabited post-industrial heaps were significantly thicker than the reference leaves. Moreover, birch trees growing on heaps were characterized by a significantly higher content of flavonols and anthocyanins in leaves and higher non-photochemical quenching. In addition, birches growing on the post-coal mine heap accumulated the concentration of Mn in their leaves, which is highly toxic for most plant species.

Increasing the thickness of the leaves, the content of flavonols and anthocyanins, and efficient non-photochemical quenching seem to be important features that improve the colonization of extreme habitats.

The study has been well conducted. However, some revisions are still required as shown below;

Comment 1: The introduction is somewhat well written, but the authors should add a separate section in the Introduction section discussing the research hypothesis in detail and you may also include some more recent literature highlighting that.

Response: We thank the Reviewer for the comment. A relevant paragraph has been added to the introduction as suggested. (See lines 85-97 in improved MS).

Comment 2:  Plant sampling is well-explained in the Methods section, but the authors should also write the methods used in more detail to be reproducible.

Response: We have detailed the description of the research methods used, according to the Reviewer's suggestion (See lines 151-172).

Comment 3: How many replicates the authors used?

Response: Ten representative trees were selected for each site. Nevertheless, the number of measurements of selected parameters was different, and so on a single tree we made fifteen measurements of the pigment content in leaves (a total of 150 measurements for each site), ten measurements of gas exchange (a total of 100 measurements for each site), but also four PAM fluorescence measurements (a total of 40 measurements for each site). We considered a single examined tree to be a biological replicate, therefore in the graphs n = 10, although behind this 10 there are always 40 to 150 real repetitions of measurements.

Comment 4: In Results section, please explain Figure 2 “Chlorophyll a fluorescence induction curves” further and highlight its significance “as the current explanation is too short!”

Response: The description of the Figure 2, both in the legend and in the Results section, has been extended, according to the Reviewer suggestion (See lines 266-291).

Comment 5: Discussion should be interpreted with the results of this work and previous work “comparison” agreement or disagreement” in more details too.

Response: We tried to highlight more details in the discussion, moreover, we expanded it with new literature (See lines 390-470).

Comment 6:  Conclusion section should also include the concluded recommendation for future work. Adjust that.

Response: Recommendations were added according to the Reviewer comment (See lines 487-492).

The manuscript was philologically corrected, linguistic and grammatical errors were removed, as suggested by the Reviewer.

Round 2

Reviewer 3 Report

The authors have revised the manuscript as per most of my suggested comments